# Shift Register, Reconvergent-Fanout (SiRF) PUF Implementation on an FPGA

**Jim Plusquellic** 

Department of Electrical and Computer Engineering, University of New Mexico, Albuquerque, NM 87131, USA; jplusq@unm.edu

**Abstract:** Physical unclonable functions (PUFs) are gaining traction as an attractive alternative to generating and storing device keying material over traditional secure non-volatile memory (NVM) technologies. In this paper, we propose an engineered delay-based PUF called the shift-register, reconvergent-fanout (SiRF) PUF, and present an analysis of the statistical quality of its bitstrings using data collected from a set of FPGAs subjected to extended industrial temperature-voltage environmental conditions. The SiRF PUF utilizes the Xilinx shift register primitive and an engineered network of logic gates that are designed to distribute signal paths over a wide region of the FPGA fabric using a MUXing scheme similar in principle to the shift-rows permutation function within the Advanced Encryption Standard algorithm. The shift register is utilized in a unique fashion to enable individual paths through a Xilinx 5-input LUT to be selected as a source of entropy by the challenge. The engineered logic gate network utilizes reconvergent-fanout as a means of adding entropy, eliminating bias and increasing uncertainty with respect to which paths are actually being timed and used in post-processing to produce the secret key or authentication bitstring. The SiRF PUF is a strong PUF build on top of a network with 10's of millions of possible paths.

**Keywords:** physical unclonable function; path delay; FPGA

## 1. Introduction

As small embedded system components of the IoT era continue to proliferate, the attack surface for adversaries becomes increasing rich with opportunities to subvert, impersonate and compromise the integrity of these systems as a means of stealing identities and sensitive information, denying customer services and/or causing system failure. The limited resources available within IoT devices, and the unsupervised wireless environments in which they are commonly deployed, increases the difficulty of creating secure defenses against attacks and maintaining trust that the device has not been manipulated or cloned with a malicious substitute.

Physical unclonable functions (PUFs) can be used to provide a hardware-based security and trust infrastructure for the device. At their core, PUFs are secure, tamper-evident bitstring generation primitives capable of providing device-specific keys for encryption algorithms (running in hardware or software on the device), or unique, one-time-usage bitstrings for challenge-response bilateral authentication protocols between a trusted server and the device. Moreover, the tight coupling between the PUF and the physical device characteristics enhances trustworthiness, making it nearly impossible for an adversary to create a clone. The PUF is therefore ideally suited for security sensitive applications running on IoT devices.

Key to the strength of PUF-generated secret keys and authentication bitstrings is a large source of entropy. Entropy refers to the average level of information contained in a message or bitstring. Messages that possess a high level of entropy lack patterns or repeated sequences, similar to a highly compressed image. PUFs derive entropy from physical imperfections in the device manufacturing process, e.g., threshold voltage and

doping concentration variations associated with transistors, line edge roughness of metal and polysilicon wires, etc. Variations in the physical and chemical characteristics of semiconductors result in changes in the electrical response characteristics of the device. Most PUFs measure some type of electrical response signal at high resolution, and extract and convert small analog signal variations into a digital bitstring.

Unfortunately, most physical imperfections are not truly random in nature and instead exhibit various forms of bias. Bias can appear in the form of design bias, where identically designed circuit features are not truly identical because, e.g., of stress effects that exist along the edges of the die. Or bias may exist in the physical properties of the manufactured material, e.g., variations in doping profiles or line width variations that are not perfectly Gaussian across regions of the chip. Bias is always disruptive to the PUF's bitstring generation process, creating non-random features such as patterns or long sequences of 0's and 1's, thereby reducing entropy. PUF architectures therefore need to remove bias, counteract its effects or avoid it altogether by using correct-by-construction techniques.

In this paper, we present a novel PUF architecture that is engineered to counteract bias and reduce sources of variations that are not truly random. The PUF utilizes shift registers and MUXs to create structural diversity in the placement and routing of a gate-level netlist, and reconvergent-fanout to add uncertainty regarding which logic gates actually define the paths whose delays are measured and utilized in the bitstring generation process. By distributing the path structures over a large region of the FPGA fabric, localized bias effects that exist in the delays of individual logic gates (realized using look-up tables or LUTs within the FPGA) act to cancel each other out. Post-processing techniques including a global process and environmental variation compensation method are utilized to significantly reduce undesirable changes in delay, e.g., delay variations introduced by changes in temperature or supply voltage, as well as delay variations that occur globally within all components on the device, a.k.a. chip-to-chip process variations.

The specific contributions of this paper are summarized as follows:

1. A novel delay-based PUF architecture that reduces undesirable bias by distributing components of the sensitized paths over a wide region of the FPGA using shifters, MUXs and logic gate networks.
2. A hazard-free by-construction logic gate netlist that leverages reconvergent-fanout to add uncertainty regarding which of the inputs of logic gates on the sensitized path dominate the path timing and determine the overall path delay.
3. A calibration method that post-processes the measured digitized path delays to reduce both environmentally-induced changes in path delay and those introduced by global process variation effects.
4. An entropy enhancing method that uniformly and randomly distributes within-die delay variations, and maximizes the number of bits available for secret keys and authentication operations.

*Background*

Most PUF architectures proposed in the literature are constructed as an array of identically-designed sub-circuit components, e.g., individual transistors [1], ring-oscillators (ROs) and/or arbiters (ARB) [2–4], latches [5], static random access memory (SRAM) cells [6], resistive [7] or capacitive [8] elements.

For example, the authors of [9] utilize the random power-up state of the flip-flops (FFs) on FPGAs as a PUF. The bitstream is modified to prevent reset of the FFs in the programmed design, and then the random power-up values are read out in a manner similar to the SRAM PUF. Majority voting schemes and fuzzy extractors are proposed to address reliability and bias issues which are observed in the FF values. A second memory-based PUF, called the Butterfly PUF, is proposed in [10], which is implemented as two cross-coupled latches with the stimulus control signal connected to the clear and preset signals of the latches.

A specialized version of the ring-oscillator (RO) PUF is proposed in [11] called the Transient Effect Ring Oscillator (TERO) PUF. The TERO PUF structure is partitioned into

two distinct, identically routed components called branches, where each branch has a 2-input AND and an enable signal. A start signal enables oscillations in each branch, which terminates to a stable state when one of the edges catches up with the other.

A delay difference PUF (DD-PUF) is proposed in [12] in which a ring is defined with two identically designed path segments, $P_1$ and $P_2$. The path segments consist of a latch followed by an inverter, where excitation is similar to the TERO PUF with oscillations occurring until one edge catches up with the second, creating a stable condition. A novel lightweight extension is proposed in [13] which cross-couples XOR gates and utilizes two series inserted AND gates to enable oscillations. The design is extremely compact, yielding 4 PUF bits per CLB in Xilinx Spartan and Artix devices.

Another compact PUF architecture, called the PicoPUF, uses a pair of FFs connected to the inputs of a cross-coupled NAND gate [14]. A 1-bit cell fits into one slice of an FPGA. Follow-on work in [15] carried out a large scale evaluation of the PicoPUF and compared metrics including uniformity, bit-aliasing, reliability, min-entropy, etc. with the RO PUF and several memory-based PUFs.

A reconfigurable version of the PicoPUF is proposed in [16]. Control signals and configurable logic structures are incorporated to allow multiple different PicoPUF instances to be created, thereby increasing the number of bits and improving the resource utilization. The configuration is accomplished using a series connected set of 2-to-1 MUXs which select one of two inverters as a delay element. The configurable logic path is inserted between the launch FFs and SR-Latch arbiters.

Notable exceptions to these identically-designed sub-circuit array constructions include the PUF architecture proposed by Huang, et al. [17], Aarested et al. [18–21] and Sauer et al. [22], which utilize arbitrary functional units. No attempt is made to constrain the lengths of the paths in these latter proposed PUF architectures and instead pre- and post-processing methods are leveraged to address undesirable path length bias effects.

Huang et al. [17] first proposed the use of arbitrary functional units as a source of entropy for PUFs. They measure path delays accurately using a set of negatively-skewed shadow registers placed on the end points of selected combinational paths and repeatedly apply the test sequence while phase shifting the shadow register clock by 160 ps decrements until the shadow register and data path registers differ. The number of phase shifts is used as the digitized path delay and directly as the response to the challenge. The responses defined in this fashion make them subject to error due to circuit hazards, measurement noise and temperature-voltage environmental variations. The proposed authentication protocol defines a ring-oscillator-based temperature compensation technique and fuzzy matching to deal with bit flip errors.

Aarested et al. [18–21] also utilize arbitrary functional units as a source of entropy in a PUF called the hardware-embedded delay PUF or HELP, but constrain logic synthesis to generate a hazard-free circuit implementation as a means of avoiding path timing inaccuracies introduced by circuit hazards. The authors also define a post-processing algorithm that compensates for the adverse effects on path delays caused by environmental changes in temperature and supply voltage, a modulus operation that reduces path length bias issues as well as a device population-based offset method for optimizing inter-chip HD. They propose session key generation and authentication protocols that leverage bit flip avoidance methods where helper data from both the device and server are utilized to further improve intra-chip HDs. The drawbacks of HELP include the complexity associated with deriving challenges, controlling the length and structure of the sensitized paths used in the bitstring generation process and the increased size of the netlist to accommodate a hazard-free implementation (using WDDL [23]).

Sauer et al. [22] propose a sensitized path (SP) PUF that uses circuit analysis methods to pre-characterize hazard-free paths through an arbitrary functional unit to identify paths of similar length. The SP PUF architecture incorporates MUXs on path end points to enable a race-resolution element (similar to an Arbiter PUF) to determine which path in a pair of simultaneously tested paths is faster. Fuzzy extraction is utilized to improve entropy

and to provide error correction. The drawbacks associated with the SP PUF are similar to those described above for HELP, with additional complexity introduced during circuit analysis pre-characterization to identify paths of similar length and with sufficient entropy. Moreover, the threshold used in the path pairing process will produce distributions that are not centered around a zero mean, and will therefore result in some level of bias during bitstring generation. Last, the PUF infrastructure introduces additional overhead for the MUXs, race-resolution elements and fuzzy extractors.

The SiRF PUF is engineered to improve further on the rich entropy source that exists within arbitrarily synthesized functional units by:

- Leveraging within-die delay variations that occur in the series combination of LUT, switches and wires within the FPGA fabric.
- Utilizing shift register paths, MUXs and reconvergent-fanout to distribute the structural components of sensitized paths over a wide region of the FPGA fabric as a means of reducing bias effects that exist within localized regions.
- Simplifying design pre-characterization, constraints and challenge generation while maintaining an exponentially large CRP space
- Enabling control over path length using a modular architecture, which provides the opportunity to tune signal-to-noise levels to meet statistical quality targets for reliability, uniqueness and randomness in the generated keys and bitstrings.

We also describe novel methods for post-processing path delay measurements into bitstrings that remove path length bias and calibrate for delay variations introduced by chip-to-chip process variations and adverse environmental conditions.

## 2. SiRF Design

A block diagram of the SiRF PUF is shown in Figure 1. The physical design is constructed as a set of modules, arranged in a set of rows and columns. For the FPGA experiments carried out in this paper, the SiRF PUF is constructed with three rows, $row_0$ through $row_2$, and eight columns, $col_0$ through $col_7$. Logic signal transitions are introduced by a set of 32 Launch FFs and path delays are measured by selecting a path end point using a 32-to-1 MUX and directing the signal to a time-to-digital converter (TDC). One of the twenty-four modules is highlighted in pink in the figure.

A module consists of a set of four XNOR gates, each of which drives the shift-clock input of a corresponding LUT. The contents of the shift register are initialized with the pattern "0101...01". A rising transition on the shift clock input introduces a second rising or falling transition within the LUT by shifting the contents of the shift register by one bit position to the right, with circular wrap as shown by the call-out in the figure. The input bits of the LUT are driven by a set of challenge bits that select one of the 32 paths through the LUT as the tested path. Even challenges, i.e., those with the low-order bit set to 0, create (0 –> 1) rising transitions while odd challenges create falling transitions. The low order bit of the LUT input challenge, labeled as transition direction challenge bit ($TC_x$), is routed to the low order bit of all LUTs on the row, i.e., across the eight modules. This ensures that the combinational logic driven by the shift register LUTs within the modules on the row remains glitch-free, as we discuss further below.

The four shift register LUT outputs within a module fan-out to drive a set of four logic gates and four 4-to-1 MUXs. The logic gates implement AND, OR and AND-OR non-inverting functions as shown in Figure 1. The outputs of the logic gates drive selected inputs to the MUXs but also fan-out and route to other modules, which distributes signal paths across the row. The output of the MUXs drive a second row of logic gates arranged in a dual network fashion to the first row, to complete a reconvergent-fanout wiring network for the row. A set of 16 challenge bits configure the two rows of MUXs within each module. Therefore each module accepts (32 + 1) input challenge bits, 17 for the shift register inputs and 16 for the eight 4-to-1 MUX inputs. The experimental design constructed and tested in this paper has eight columns and three rows and therefore has an input challenge size equal to $24 \times 33 = 771$ bits.

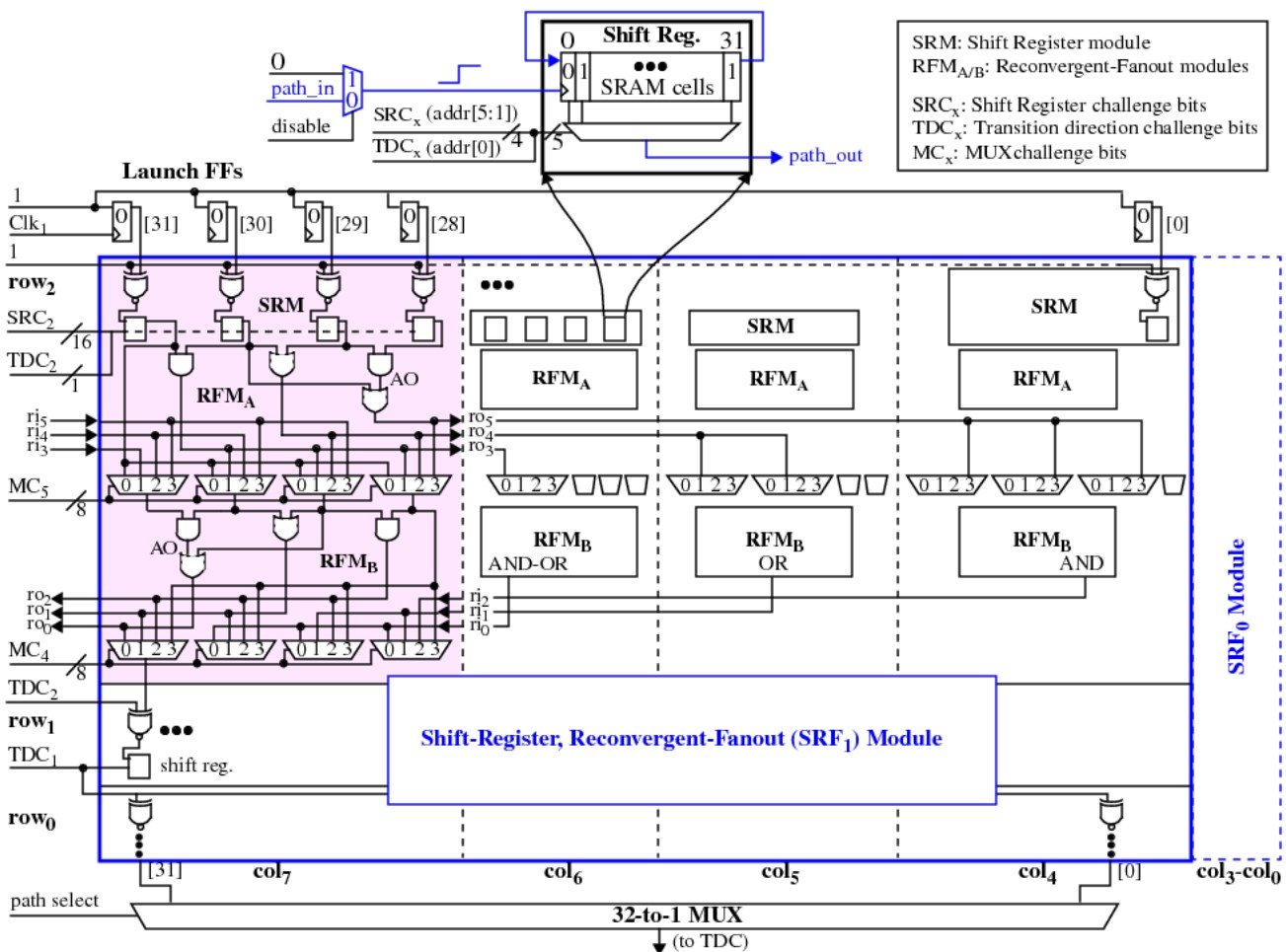

**Figure 1.** Block diagram of the shift-register, reconvergent-fanout (SiRF) PUF.

The signal transitions on the outputs of each module can be rising or falling. However, the shift register clock inputs of the next module must be driven with a rising transition. The module outputs are inverted if needed using the $TC_x$ inputs, which fanout to one of the inputs of all XNOR gates on the row. Therefore, when the $TC_x$ challenge bit is set to 1 for the row, the modules in the row test paths through the shift register LUTs with falling transitions. This same $TC_x$ bit also drives the XNOR gate inputs of the following module to invert the falling transitions on the module outputs to rising transitions, which is required by the shift registers of the following module. The signal constraints as described are sufficient to ensure every path through the SiRF netlist is hazard-free by construction.

It is important to note that the SiRF netlist components do not require any placement or routing constraints, as is true for many of the proposed identically-designed subcircuit-based PUFs. The SiRF PUF addresses the disruptive effects of bias using SpreadFactors, which are described in the following.

## 3. The SiRF PUF Algorithm

The SiRF PUF algorithm generates nonces, authentication bitstrings and encryption keys by applying a series of mathematical operations to sets of digitized path delays (DV). The DV are obtained by applying challenge vectors $Chlng_a\{v\}$ to the SiRF netlist inputs shown in Figure 2. The challenge vectors reconfigure the logic gate network and define a unique set of paths.

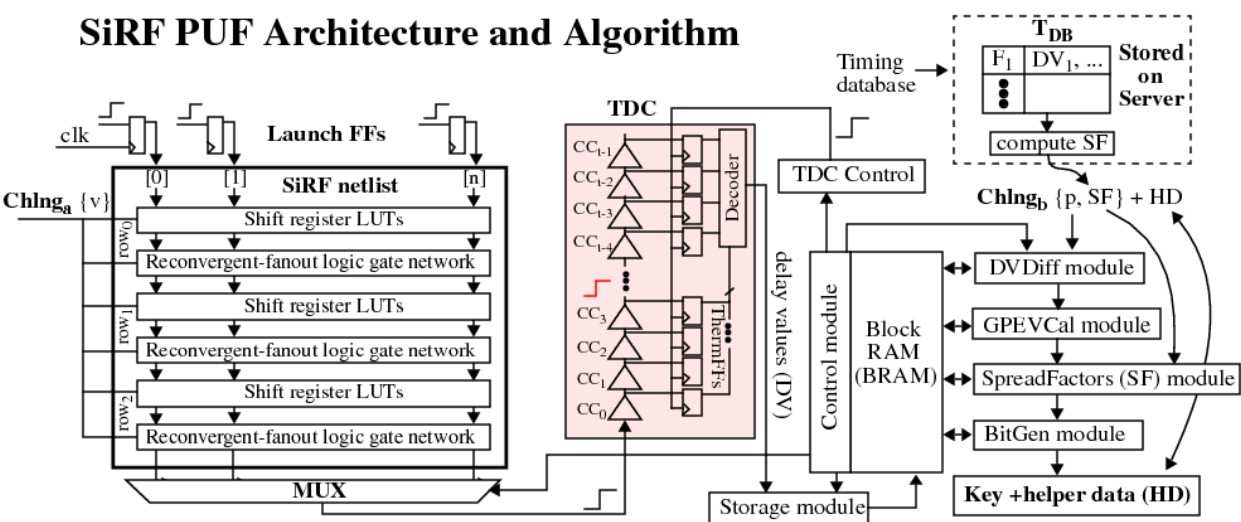

**Figure 2.** Shift-register, reconvergent-fanout (SiRF) PUF architecture (**left**) and modules defining the algorithm (**right**).

The delays of the paths are measured one-at-a-time using a time-to-digital converter (TDC). The TDC utilizes a fast carry-chain to digitize the analog delay of a path into an integer value, which represents the number of carry-chain-buffer $\Delta$-t delays that are required for the signal propagating along a path to emerge on the MUX output of the SiRF netlist and propagate into the carry chain of the TDC [24]. A user parameter specifies the number of samples to average to obtain a 16-bit fixed point representation of each DV, with 4 bits of binary precision.

The fixed point DV are stored in a Block RAM (BRAM) for subsequent processing into bitstrings and keys by the modules shown in the control flow diagram on the right side of Figure 2. The SiRF PUF algorithm is currently configured to generate a set of 4096 DV, which are used, along with a set of user-defined parameters $p$ and ancillary data SF, as the inputs to the modules. The latter components define a second component of the challenge, $\text{Chlng}_b\{p, \text{SF}\}$. The HD component in the figure refers to helper data, which is an output of the BitGen module during enrollment and is an input during regeneration.

Each of the modules performs an operation designed to improve one or more of the statistical properties of the generated keys and bitstrings, and in some cases, expands the challenge-response space of the PUF, as discussed further in the following.

### 3.1. Difference Module

The Difference module partitions the 4096 DV into two classes, rising and falling. The rising class ($\text{DV}_R$) refers to paths that propagate a 0-to-1 transition in the first row of the SiRF netlist, while paths in the falling class ($\text{DV}_F$) propagate a 1-to-0 transition. Figure 3 plots the $\text{DV}_R$ and $\text{DV}_F$ distributions collected from an FPGA $F_1$ under nominal temperature and supply voltage (TV) conditions, given as $\{25\,^{\circ}\text{C}, 1.00\,\text{V}\}$. The Difference module runs two primitive 11-bit LFSRs to select pseudo-randomly from these distributions, using two user specified seeds, to create pairwise differences called DVD, as shown on the right side of the figure. The two seeds uniquely pair all 2048 $\text{DV}_R$ with all 2048 $\text{DV}_F$. The Difference module expands the challenge-response space of the PUF to $2^{22}$ bits via the two 11-bit LFSR seeds.

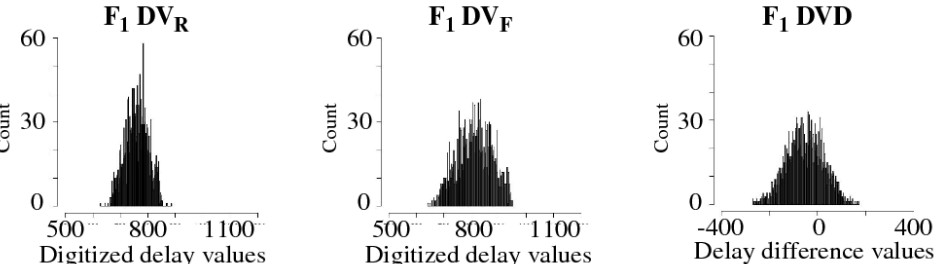

**Figure 3.** DVD operation of the SiRF PUF algorithm illustrating $DV_R$ and $DV_F$ histogram distributions, and the corresponding DVD distribution for FPGA $F_1$.

### 3.2. GPEVCal Module

The DVD distribution for an FPGA will contract and expand as environmental conditions change. Moreover, chip-to-chip performance differences in the FPGAs produce DVD distributions of various widths even when tested under the same environmental conditions. Chip-to-chip performance differences represent a form of entropy for the PUF, but unfortunately, they must be removed because of the correlation they introduce in the bitstrings and keys, which adversely impacts the uniqueness and randomness properties.

The GPEVCal module performs two linear transformations to address these issues. The first one normalizes the DVD using Equation (1). The mean and range of the distribution are computed using Equations (2) and (3), respectively, where |DVD| represents the cardinality of the DVD (which is 2048 in this analysis), and $N$ represents the normalized distribution. The second transformation simply expands the $N$ distribution around 0 by multiplying $N$ by a user-specified range constant $r_c$ using Equation (4). The impact of these transformations is shown by the $F_1$ DVD and $F_1$ DVD$_c$ distributions in Figure 4, with $r_c$ specified as 128.

$$N = \frac{(\text{DVD} - \mu)}{\text{range}} \tag{1}$$

$$\mu = \frac{\sum_{j=1}^{|\text{DVD}|} \text{DVD}_j}{|\text{DVD}|} \tag{2}$$

$$\text{range} = \max_{\forall j \in |\text{DVD}|} \text{DVD}_j - \min_{\forall j \in |\text{DVD}|} \text{DVD}_j \tag{3}$$

$$\text{DVD}_c = N \times r_c \tag{4}$$

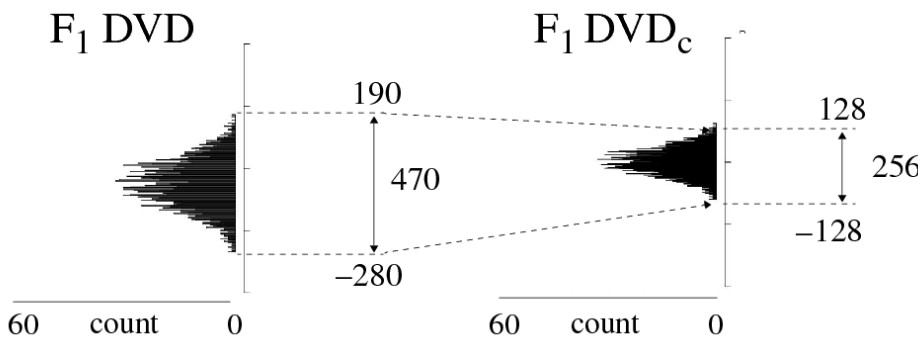

**Figure 4.** GPEVCal module operation showing the transformation of the DVD distribution to DVD$_c$ for FPGA $F_1$.

### 3.3. SpreadFactors Module

The SpreadFactors module is tasked with optimizing the uniqueness and reliability-based statistical properties of the SiRF PUF's keys and bitstrings and with maximizing the number of stable bits that are extracted by the BitGen module. SpreadFactors are used to address a source of bias that does not exist in identically-designed PUF architectures, namely, bias introduced by path length differences inherent in the design of the SiRF netlist. As mentioned earlier, there are no place-and-route constraints associated with the SiRF netlist component, and therefore, the $DVD_c$ created by the GPEV module cannot be assigned a bit value until the path length differences (bias) is removed.

Design bias is removed by first carrying out a characterization process using the DV collected from a sample of devices in the population. The characterization process is performed by a server that securely stores the DV in a timing database, labeled $T_{DB}$ in Figure 2. Once the challenge is generated for a device, the server selects sets of DV from the data stored in $T_{DB}$ and then applies the Difference and GPEVCal module operations (in software) to produce a set of DVD and $DVD_c$ distributions, respectively, for each of the chips in the sample. The DVD and $DVD_c$ distributions for device $F_1$ are shown in Figure 4 as an example from the set stored in the $T_{DB}$.

Once the $DVD_c$ distributions are created, a SpreadFactor $SF_i$ is then computed for each $DVD_{c_i}$ as the median value of the set $DVD_{c_{i,j}}$ for all devices $j$ in the sample, as given by Equation (5). The server transmits the SpreadFactors, $SF_i$, to the device. The device applies the challenge to its hardware instance of the PUF to derive the set $DVD_c$, and subtracts the $SF_i$ from each $DVD_{c_i}$ to remove the design bias.

$$SF_i = \text{median}_{\forall j \in |\text{devices}|} DVD_{c_{i,j}} \tag{5}$$

The SpreadFactors can be sent unencrypted to the device because they reflect only the median values of the chip sample and reveal no information about the relative positions of the $DVD_{c_i}$ generated by the device. The SpreadFactors ensure each set of $DVD_{c_i}$ are split evenly across the 0 mean, and act to improve inter-chip hamming distance ($HD_{\text{inter}}$) assuming the values stored in $T_{DB}$ represent the distribution charateristics of the population.

As an example, the curves in Figure 5a show the transformations that occur for $DVD_1$ as the GPEVCal and SpreadFactor operations are applied. Each curve corresponds to one FPGA from a set of 120. The large variations in the $DVD_1$ on the left are introduced by performance differences in the FPGAs. As discussed earlier, this source of entropy introduces correlations, i.e., long strings of '0' s and '1' s in the bitstrings and keys, and must be removed.

GPEVCal removes chip-to-chip performance differences and collapses the distribution as shown by the $DVD_{c1}$ in Figure 5a. A large portion of the variation that remains is referred to as within-die variations or WID. Although WID measures delay variations that occur across devices, it also models the variation that would exist if identical copies of the paths were tested on the same device. This is true because GPEVCal removes chip-to-chip variations and therefore, each of the $DVD_{c1_i}$ possess the same performance characteristics.

As indicated earlier, GPEVCal also addresses variation introduced by adverse environmental conditions. Figure 5b plots a set of temperature-voltage curves corresponding to one of the devices from Figure 5a. The curves are color-coded according to the temperature conditions, with curves of the same color corresponding to different supply voltages. The impact of the environment on delay exceeds 25 delay units as shown by the $DVD_1$ points, but is significantly reduced by GPEVCal in the $DVD_{c1}$ points. The variation that remains is referred to as uncompensated TVNoise (UC-TVN).

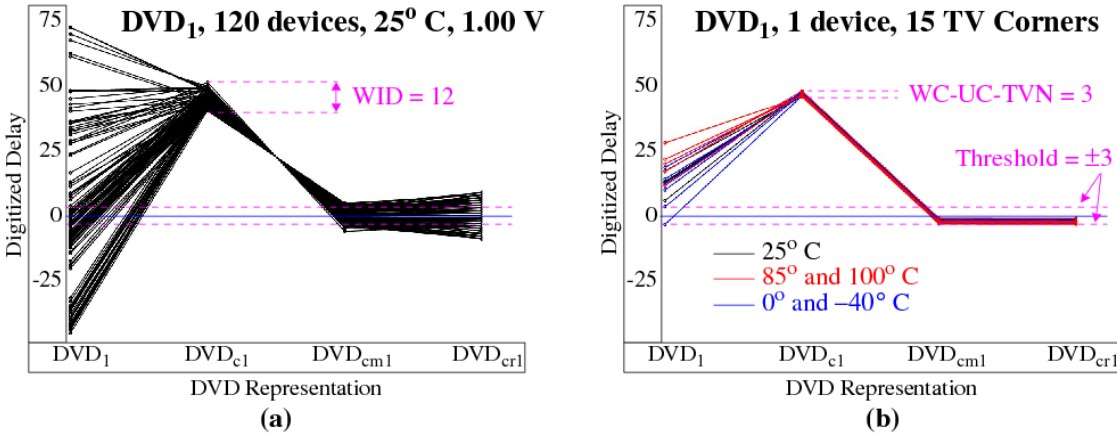

**Figure 5.** Transformations carried out by the GPEVCal and SpreadFactor modules showing progression of $DVD_1$ to $DVD_{c1}$, $DVD_{cm1}$ and $DVD_{cr1}$ for all FPGAs under nominal conditions (**a**) and for one FPGA under all TV corner conditions (**b**).

WC-UC-TVN is defined as the largest difference (worst case) between a $DVD_{ci}$ for one of the TV corners and the $DVD_{ci}$ measured under nominal conditions (NC). The WID and WC-UC-TVN, which are given as 12 and ±3 in Figure 5a,b, are computed using Equations (6) and (7), respectively. The WID and WC-UC-TVN for the 2048 $DVD_{ci}$ computed for a SiRF PUF challenge are plotted in Figure 6. The WID are larger than the WC-UC-TVN in all cases, e.g., the largest WC-UC-TVN is 5.2 while the smallest WID is 6.0. This indicates that the entropy, represented by WID, is always larger than the noise floor given by WC-UC-TVN. The mean WID is 11.1, which is well above the mean WC-UC-TVN of 2.7. The WID versus WC-UC-TVN analysis is useful for providing guidance on selecting the threshold parameter used by the BitGen module discussed below.

$$\text{WID}_{\text{DVD}_{c_i}} = \max_{\forall j \in |\text{devices}|} \text{DVD}_{c_{i,j}} - \min_{\forall j \in |\text{devices}|} \text{DVD}_{c_{i,j}} \tag{6}$$

$$\text{WC-UC-TVN}_{\text{DVD}_{c_i}} = \max_{\forall k \in |\text{TV corners}|} abs(\text{DVD}_{c_{i,k}} - \text{DVD}_{c_{i,NC}}) \tag{7}$$

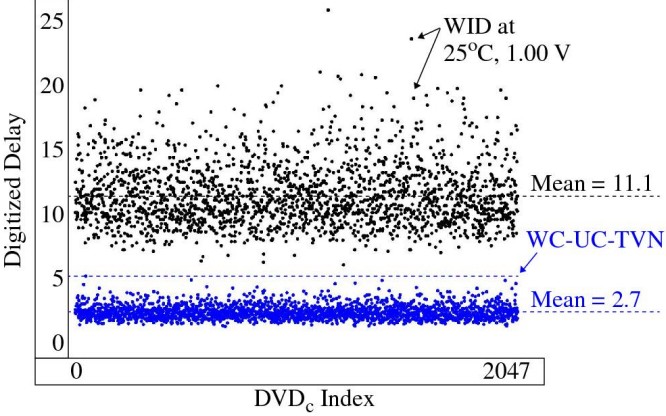

**Figure 6.** WID and WC-UC-TVN for all 2048 $DVD_{ci}$ using 120 devices.

The median value of the $DVD_{c1}$ distribution computed by the server is used to shift the distribution and center it over 0, as shown by the $DVD_{cm1}$ in Figure 5a. The SpreadFactors module also flips the distribution (as shown in the figure) if the magnitude of the median value falls within any one of a set of user-defined regions above or below 0. The flip operation increases randomness by factoring in other unknowns, in particular, the length of the tested paths in the SiRF netlist.

The input to the BitGen module can be the $DVD_{cm}$, or the SpreadFactors module on the server can optionally carry out a second randomization operation that helps to increase the number of bits classified as stable (and useable) during the BitGen operation. The distribution associated with each $DVD_{cmi}$ is not uniform, and is typically Gaussian-like in shape as shown for $DVD_{cm1}$ on the left side of Figure 7. The high concentration of the $DVD_{cm1}$ around 0 will result in many of the devices discarding this $DVD_{cm1}$ as unstable during bitstring generation.

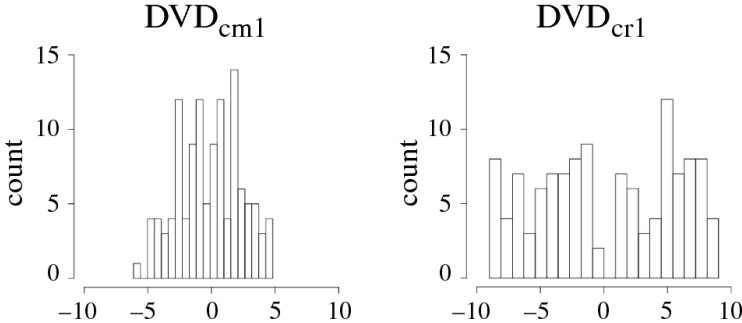

**Figure 7.** Distributions associated with $DVD_{cm}$ and $DVD_{cr}$ from Figure 5.

The randomization operation distributes the $DVD_{cm1}$ uniformly by adding a small random value to the $SF_i$. The resulting distribution for $DVD_{cr1}$ is shown on the right side of Figure 7, which for larger sample sizes will become increasingly uniform. The randomization process does not change the bit value associated with the $DVD_{cm1}$, i.e., the $DVD_{cm1}$ that are greater than 0 will remain greater than 0 after randomization and vise versa. The points labeled $DVD_{cr1}$ in Figure 5a show the final distribution characteristics.

*3.4. BitGen Module*

The BitGen module uses a user-defined threshold parameter to classify $DVD_{cr}$ as weak or strong. The threshold is tunable with larger values providing improved reliability during regeneration of the response bitstring under adverse operational conditions. The $DVD_{cr}$ are first partitioned into two regions by the horizontal line at 0, with values above the line assigned a bit value of 1 and values below the line assigned 0. The threshold further partitions each group of $DVD_{cr}$ into strong and weak classes.

As an example, a subset of the $DVD_{cr}$ for device $F_1$ are plotted along the x-axis in Figure 8. The term enrollment is used in reference to the process of measuring the DV for the first time, typically under nominal environmental conditions. The two horizontal dotted lines in Figure 8a represent the thresholds, which are placed at equal-distance positive and negative displacements around 0. Only the $DVD_{cr}$ classified as strong are shown in the figure.

The $DVD_{cr}$ within the threshold regions are closer to 0 and are classified as weak. Weak $DVD_{cr}$ are less reliable, i.e., they have a higher probability of producing bit flip errors during regeneration, because UC-TVNoise can displace the $DVD_{cr}$ across the 0 line. In other words, regenerated $DVD_{cr}$ within the threshold region have less tolerance to change. Consequently, weak $DVD_{cr}$ are excluded from the bitstring generation process by labeling them with a 0 in the helper data bitstring (discussed below). Strong $DVD_{cr}$, on the other hand, are located above the upper threshold or below the lower threshold, and are labeled with a 1 in the helper data bitstring. Thresholding improves the reliability of the bitstring regeneration process by creating a buffer against uncompensated environmental disturbances. Thresholding also reduces the original length of the bitstring from 2048 to 1605 as shown.

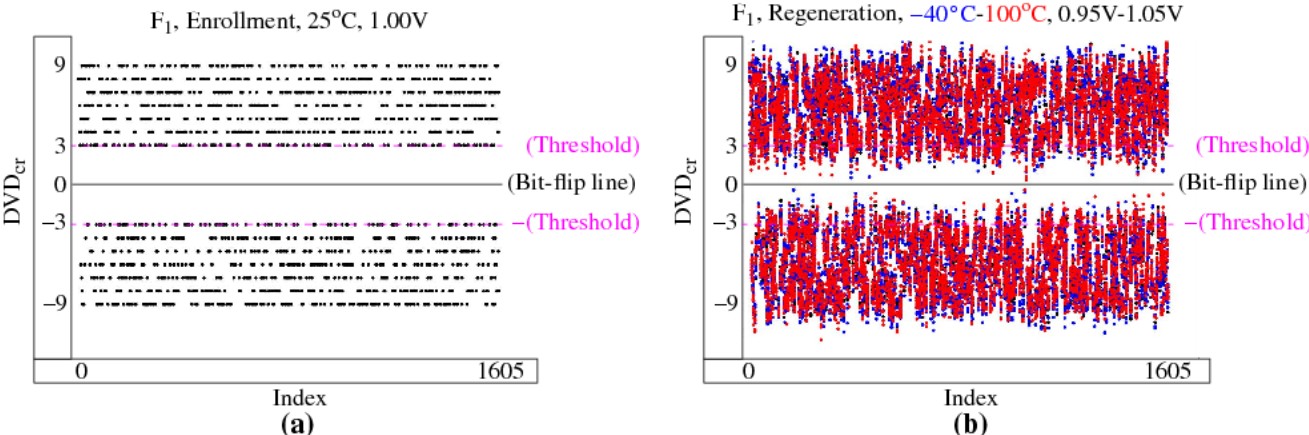

**Figure 8.** $DVD_{cr}$ for FPGA $F_1$ computed under nominal conditions during enrollment (**a**) and under a set of TV corners during regeneration (**b**). Only the strong bits are shown using a threshold of 3.

The $DVD_{cr}$ computed for device $F_1$ at each of the TV corners are plotted in Figure 8b. The TV corners used in our experiments are defined as all possible combinations of temperatures $25\ ^\circ C, -40\ ^\circ C, 0\ ^\circ C, 85\ ^\circ C, 100\ ^\circ C$ and voltages $0.95\ V, 1.00\ V, 1.05\ V$, and are color-coded in the figure according to temperature. The term regeneration is used in reference to the process of re-measuring the DV identified as stable during enrollment, potentially under worst case environmental conditions.

The encroachment of the red and blue data points within the threshold region towards 0 graphically portrays the adverse effects of UC-TVNoise. Although no bit flip errors occur in this data set instance, several data points are very close to the bit flip line. In order to achieve high levels of reliability, e.g., less than one bit flip error in a million, a larger threshold may be required.

### 3.4.1. XMR Reliability Enhancement

XMR refers to another technique for improving reliability, which can be used as an alternative to increasing the threshold. XMR refers to a suite of methods that utilize redundancy to prevent bit flip errors during regeneration. The 'X' in XMR refers to the level of redundancy, with familiar variants such as triple-modular redundancy (TMR) providing the lowest level of protection with three redundant copies, and 5MR increasing the level of protection to five redundant copies, etc.

The entropy inherent within the SiRF PUF can be used to generate the response bitstring with XMR or the response bitstring can be used as input to XMR. In the latter scenario, the response bitstring can refer to some type of confidential information, e.g., passwords or bank account numbers. Here, XMR is tasked with both encoding the confidential information and adding reliability to enable its error-free regeneration. We refer to the former mode of XMR as first-strong-bit (FSB) [25] and the latter mode as secure-key-encoding (SKE) [26]. In this paper, we focus our analysis on FSB mode and leave the SKE mode analysis for a future work.

An example application of TMR is shown in Figure 9 using the $DVD_{cr}$ generated by device $F_1$. Enrollment is performed under nominal conditions while regeneration is carried out under the remaining fourteen TV corners. The enrollment $DVD_{cr}$ are shown as a thick black curve in the figure, while the regeneration curves are color-coded as red (elevated temperature) and blue (cold temperature).

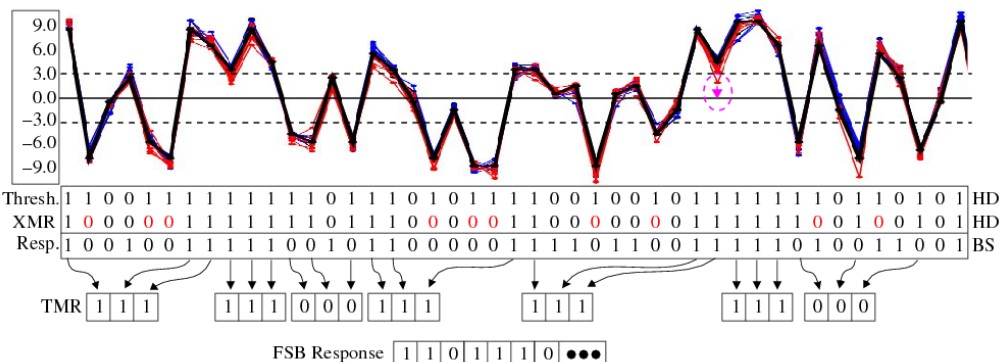

**Figure 9.** XMR FSB mode example application using DVD$_{cr}$ generated by device $F_1$ under nominal conditions (thick black line) and under a set of fourteen TV corners.

The Thesholding operation carried out during enrollment uses a threshold of 3 and creates the helper data (HD) bitstring labeled as 'Thresh.' in the figure. The bit positions labeled '0' indicate DVD$_{cr}$ elements that fall within the weak region during enrollment and are therefore discarded. The Thresholding HD is copied to the XMR HD initially and is updated incrementally with additional '0' s (colored red) as the XMR process is applied to the DVD$_{cr}$.

The XMR operation parses the strong DVD$_{cr}$ from left to right to construct the TMR 3-bit sequences. For example, the leftmost DVD$_{cr}$ is a strong '1', which becomes the first strong bit of the TMR 3-bit sequence. The DVD$_{cr}$ for the second bit position is a strong '0', which cannot be used as a bit in the TMR 3-bit sequence. XMR overwrites the XMR HD bit value with a '0' to exclude it and proceeds to bit positions 7 and 8 to complete the 3-bit sequence. After the third bit of the sequence is processed, a '1' is added to the FSB Response bitstring shown along the bottom of the figure. The next strong bit encountered at bit position 9 is a strong '1' and the same process is repeated to construct the next 3-bit sequence. We refer to the FSB Response bits as super-strong bits to distinguish them from the bits classified as strong by the Thresholding operation.

Regeneration uses the XMR HD as input to re-create the FSB Response bitstring. In the example shown, none of the DVD$_{cr}$ measured under the TV corner conditions creates a bit flip error. The DVD$_{cr}$ annotated with a dotted magenta circle represents the closest instance of a bit flip error. If the regenerated DVD$_{cr}$ appeared below the 0 line, then a '0' would have been generated instead of a '1' in the 3-bit TMR sequence. However, XMR would still generate a '1' in the FSB Response bitstring despite the bit flip error because majority vote is used during regeneration to determine the final FSB Response bit.

### 3.4.2. XMR Applications

The SiRF PUF and algorithm can be used in secure boot processes to generate a key-encryption-key (KEK), or it can be used to generate one-time-use authentication bitstrings and session keys for securing client-server communications. KEK applications usually require a key that is used for extended periods of time. We refer to such keys as long-lived keys (LLK) in this paper. Given that LLKs need to be regenerated many times, potentially over the lifetime of the device, reliability is an important statistical property to evaluate. Therefore we present an extensive reliability analysis using several XMR schemes in the experimental results section. Application of the SiRF PUF to securing client-server communication will be covered in future works.

### 3.4.3. DV Characterization and Challenge Selection

The goal of the challenge selection process for LLK is to select sets of configuration vectors, labeled Chlng$_a\{v\}$ on the left side of Figure 2, such that the probability of an error in the LLK key regeneration process is no greater than $10^{-6}$, i.e., one chance in a million, and ideally is much smaller, e.g., in the range of $10^{-9}$. In hardware experiments carried out

during the development of the SiRF PUF, we developed a DV Characterization process that improves further on the reliability provided by GPEVCal, Thresholding and XMR.

The objectives of the DV Characterization process are two fold. The initial phase is tasked with selecting a unique set of paths. A path is considered unique if at least one gate (LUT) is unique in the series-connected set of LUTs that define the path. The output of the first phase is a set of Characterization Vectors that are used in device experiments. The Characterization Vectors are applied to a small set of devices subjected to TV corner conditions, and a set of DV are collected and analyzed. The analysis identifies sets of compatible paths that can be used as a challenge ($\text{Chlng}_a\{v\}$), where compatibility is defined as paths whose delay changes linearly, or within a small margin of linear, as a function of TV conditions. The linearity restriction enables the linear transformation operation in the GPEVCal module to perform well across the full range of TV corner conditions.

The DV Characterization process is described as follows in reference to Figure 10. Note that the SiRF netlist is constructed to be glitch-free, i.e., with no circuit hazards, which is required for this characterization process to be effective.

1.  Behavioral VHDL describing the SiRF PUF is processed into a hard macro using the Xilinx Vivado CAD tool flow. No placement or routing constaints are utilized.
2.  A SiRF netlist and standard delay format (SDF) file is extracted from the hard macro.
3.  All possible paths through the netlist are enumerated using a C program and subsets of these paths are used as input to another C program, which determines the configuration vectors required to sensitize (test) the paths in the selected subsets.
4.  Timing simulations are carried out using the configuration vectors to produce value-change-dump (VCD) files. VCD files record the switching (transition) activity on all nodes within the netlist as the configuration vectors are applied.
5.  The VCD files and netlist are analyzed by a third C program to create a list of paths that are sensitized using the configuration vectors (Path List). Note that each configuration vector sensitizes multiple paths and it is likely that any given path may be tested more than once by the vectors.
6.  The uniquely sensitized paths from the Path List are identified using a fourth C program and the configuration vectors needed to test them are identified. The configuration vectors produced by this process are called Characterization Vectors.
7.  The Characterization Vectors are applied to a sample of at least 30 devices at each of the TV corners. The testing process produces DV files for the target paths.
8.  The DV are analyzed by a fifth C program, which selects sets of compatible paths characterized as having upper and lower bounds on UC-TVNoise and WID, respectively.

The most important, and complex, component of the DV Characterization process is operation performed in the last step, which is referred to as Challenge Vector Selection in Figure 10. The C program that implements the DV Characterization process produces a set of Configuration Vectors and Path Select Masks that test exactly 4096 paths, all of which are classified as compatible. Path compatibility is determined using an iterative refinement process which begins with a larger set of DV, each measured under a set of TV corner conditions. For example, assume a set of 30 devices are tested at 9 TV corners defined as all combinations of $\{-40\,^\circ\text{C}, 25\,^\circ\text{C}, 100\,^\circ\text{C}\}$ and $\{0.95\,\text{V}, 1.00\,\text{V}, 1.05\,\text{V}\}$, and 64,000 DV are collected for each device at each TV corner. Then the iterative refinement process for this test case is carried out using an initial set of 17,280,000 DV.

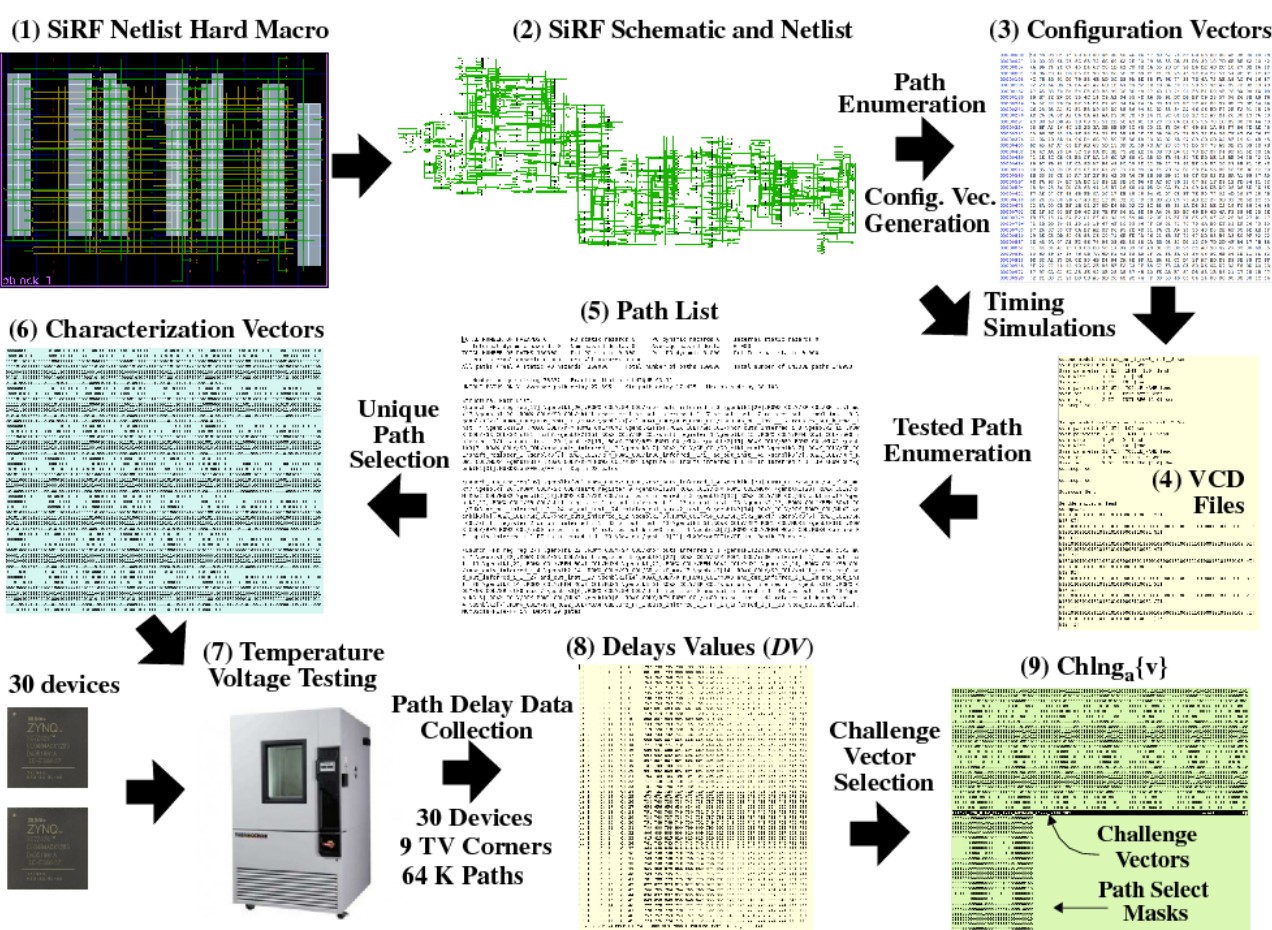

**Figure 10.** Process flow to select challenge vector sets.

On each iteration of the iterative refinement process, DV Characterization selects a subset of DV that meet or exceed two user specified limits, one that upper bounds the average level of UC-TVNoise (Ave-UC-TVNoise) and one that lower bounds the average level of WID (Ave-WID). As discussed earlier, UC-TVNoise refers to the uncorrected variation within the DV that is not removed by the GPEVCal module. Minimizing the impact of UC-TVNoise is one of the goals of DV Characterization because it acts to reduce the reliability of bitstring regeneration.

DV Characterization first applies GPEVCal to the path distributions for each chip at each TV corner. The path distributions consist of 64,000 DV per TV corner on the first iteration of the refinement process, and decrease incrementally in subsequent iterations. The expression used to compute Ave-UC-TVNoise for each $DV_x$ is given by Equation (8). The term $\mu_{DV_{x,i}}$ is the average $DV_x$ for device $i$ measured across all TV corners. The inner sum is the absolute deviation of $DV_x$ from its average value, i.e., the sum of the UC-TVNoise values for device $i$. The outer sum represents the sum of the UC-TVNoise values across all devices, while Ave-UC-TVNoise$_{DV_x}$ is its average value.

Note that the GPEVCal operation minimizes UC-TVNoise and therefore, the ideal value of Ave-UC-TVNoise$_{DV_x}$ is 0. Each of the $DV_x$ becomes a candidate to be included in the selected subset during iterative refinement if Ave-UC-TVNoise$_{DV_x}$ is less than a user specified limit. Otherwise it is removed from the set processed in subsequent iterations.

$$\text{Ave-UC-TVNoise}_{DV_x} = \frac{\sum\limits_{i=1}^{|\text{devices}|} \sum\limits_{j=1}^{|\text{TV corners}|} abs(\mu_{DV_{x,i}} - DV_{x,i,j})}{|\text{devices}| \times |\text{TV corners}|} \qquad (8)$$

The same calculation is made for Ave-WID except the inner and outer loops are reversed, as given by Equation (9). WID represents delay variations that occur for the same path on different devices and therefore the average delay across all devices, $\mu_{\mathrm{DV}_x,i}$, is used as reference point. The GPEVCal operation reduces chip-to-chip variations as well as TVNoise and therefore, the DV for all devices should be very similar except for process variations effects and small levels of UC-TVNoise. WID captures the level of entropy associated with the $\mathrm{DV}_x$, where larger values act to improve the statistical properties of the PUF. Similar to the Ave-UC-TVNoise metric, $\mathrm{DV}_x$ with values for Ave-WID$_{\mathrm{DV}_x}$ that exceed a second user specified limit are considered candidates for inclusion in the selected subset.

$$\text{Ave-WID}_{\mathrm{DV}_x} = \frac{\sum\limits_{i=1}^{|\text{TV corners}|} \sum\limits_{j=1}^{|\text{devices}|} abs(\mu_{\mathrm{DV}_x,i} - \mathrm{DV}_{x,i,j})}{|\text{devices}| \times |\text{TV corners}|} \tag{9}$$

The iterative refinement loop within DV Characterization includes a $\mathrm{DV}_x$ in the selected subset only if both limits are satisfied, otherwise it is removed from consideration. The iterative refinement loop terminates once the selected subset becomes stable from one iteration to the next, i.e., the size and composition of the $\mathrm{DV}_x$ set remains constant over two consecutive iterations.

The GPEVCal process is applied to the selected set during each iteration before the screening operation is carried out. Therefore, $\mathrm{DV}_x$ in the final selected set are compatible with each other and can be used in different combinations to define a challenge. Challenge generation randomly chooses a set of 2048 $\mathrm{DV}_R$ and 2048 $\mathrm{DV}_F$ from the compatibility set. Although the size of the compatibility set depends on the TVNoise and WID parameters specified by the user, it is typically in the range of 20,000 elements, with approx. equal numbers of $\mathrm{DV}_R$ and $\mathrm{DV}_F$.

## 4. Experimental Results

In this section, we present statistical test results that portray how well the SiRF PUF performs on meeting important cryptographic properties related to entropy, min-entropy, randomness, uniqueness and reliability. The FSB response bitstrings generated by the SiRF PUF are also subjected to the NIST statistical test suite [27]. The experimental results are derived from data collected from 120 SiRF PUF instances on Xilinx Zynq 7020 devices [28]. A challenge is derived from the compatibility set defined by the DV Characterization process from Section 3.4.3. The challenge is applied to the FPGAs to obtain sets of 4096 $\mathrm{DV}_x$ at a set of TV corners, which includes all combinations of temperatures {25 °C, 0 °C, −40 °C, 85 °C, 100 °C} and voltages {1.00 V, 0.95 V, 1.05 V}. The nominal data collected at {25 °C, 1.00 V} is used as the enrollment data in the evaluations.

The SiRF PUF parameters discussed in Section 3 include two 11-bit LFSR seeds for pseudo-randomly selecting pairs the 2048 $\mathrm{DV}_R$ and $\mathrm{DV}_F$. In total, there are $2048^2$ ($2^{22}$) unique ways of subtracting the $\mathrm{DV}_F$ from the $\mathrm{DV}_R$ to produce DVD. The statistical tests are applied to the concatenated bitstrings generated by all possible 2048 LFSR seed combinations, whose length is always less than $2^{22}$ because of the filtering carried out by the Thresholding and XMR operations.

### 4.1. Entropy and Min-Entropy Analysis

Entropy measures the average level of information contained in the FSB Response bitstrings, while min-entropy represents the worst case or the most conservative estimate of the level of information. PUF bitstring information is random, and therefore, entropy and min-entropy can also be interpreted as the level of uncertainty in predicting the outcome. The ideal value for both entropy and min-entropy is 1.0, which indicates each bit possesses a full bit of entropy, yielding bitstrings that are completely random and unpredictable.

The bar graphs in Figure 11a,b plot the average values of entropy and min-entropy for the XMR bitstrings generated for thresholds 3 and 4 and for XMR values between 3 and

11, respectively. Entropy is computed using Equation (10) while min-entropy is computed using Equation (11). The $p_i$ represent the frequency of occurrence of the two events, '0' and '1', in the full length bitstrings, while $p_{max}$ represents the larger of these two frequencies.

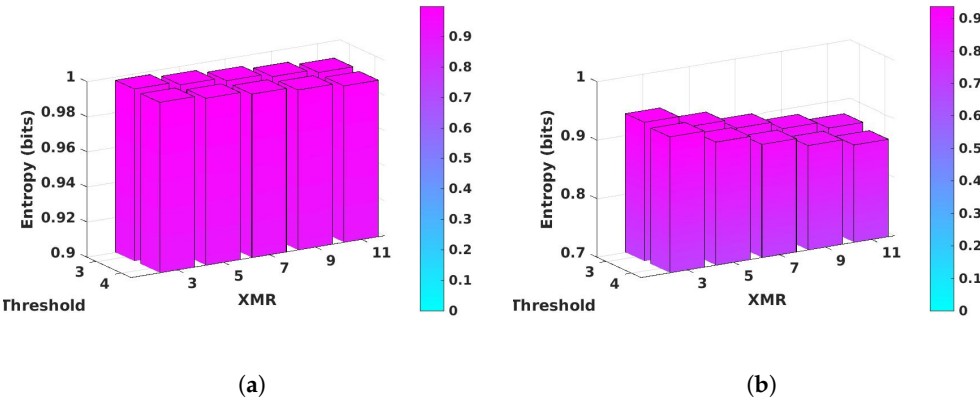

(**a**) (**b**)

**Figure 11.** Entropy results for the SiRF PUF. (**a**) Entropy of XMR bitstrings for thresholds 3 and 4. (**b**) minEntropy of XMR bitstrings for thresholds 3 and 4.

The largest values for $H(x)$ and $H_\infty(x)$ in the figures are 0.998 and 0.932 for threshold 3 and XMR 3, while the smallest values are 0.989 and 0.866 for threshold 4 and XMR 11, respectively. To the best of our knowledge, the results presented here for the SiRF PUF meet or exceed those published for other PUFs, and overall, these results indicate the bitstrings are high quality and meet cryptographic quality standards.

$$H(x) = \sum_{i=1}^{n} -(p_i \times log_2(p_i)) \tag{10}$$

$$H_\infty(x) = -log_2(p_{max}) \tag{11}$$

*4.2. NIST Statistical Test Results*

Entropy measures only one (important) aspect of random bitstrings. Assessing the level of randomness is in fact a multi-dimensional problem, and no suite of tests exist that are deemed complete. The test suite described within NIST's Special Publication 800-22 has emerged as an industry standard for evaluating PUF bitstrings [27]. The test suite was developed for evaluating true random number generators (TRNGs) and pseudo-random number generators (PRNGs), and therefore, several of the tests in the suite of fifteen expect a large number of bits. However, smaller bitstring sizes are accepted by all of the tests. If the test can be carried out successfully using the shorter bitstrings, the test output is non-zero for random bitstrings, otherwise, it is zero. The NIST test called Universal requires bitstrings with at least one million bits and is therefore excluded in our evaluation.

Each of the tests in the suite evaluates each bitstring by computing a test statistic and comparing it to a critical value that is derived from a truly random reference distribution. If the test statistic exceeds the critical value (called *alpha*), the bitstring fails the test. This happens when the test detects a pattern or anomaly in the sequence of bits based on the target criteria. All of the tests have a built-in tolerance to anomalies, which allows a small fraction of the bitstrings to fail the test while still considering the overall test passed.

The results of the NIST test suite evaluation are shown in Figure 12 for the bitstrings generated with threshold 4 (similar results are obtained for threshold 3). The x-axis gives the NIST test name while the y-axis lists the XMR values. To maintain an apples-to-apples comparison, the bitstrings are truncated to 125,000 bits for all NIST tests, which is approximately the size of the bitstrings generated at XMR 11. The default value of 0.01 is used for *alpha* and the built-in tolerance is 115, i.e., at least 115 of the bitstrings must pass each test for the overall test to be considered a pass. The z-axis plots the pass fraction,

where bars above the red dotted line at 0.958 are considered passes. All of the bars exceed the pass limit, which indicates that the bitstrings pass the NIST tests at all XMR levels.

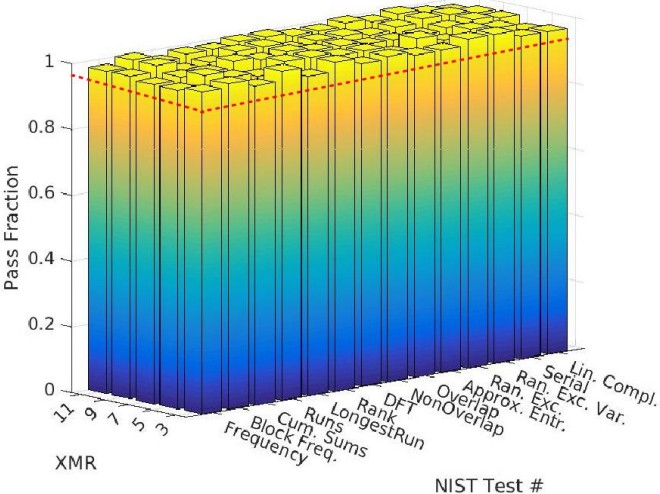

**Figure 12.** NIST statistical test results of XMR bitstrings for threshold 4.

### 4.3. Inter-Chip Hamming Distance Analysis

Inter-chip hamming distance (Inter-HD) is the common metric to measure the level of uniqueness among the bitstrings generated by a set of devices using the same challenge. Inter-HD is computed by pairing bitstrings under all combinations and then counting the number of bits that differ in each pairing. The best result occurs when half of the bits in any pairing of bitstrings differ.

$$\text{Inter-HD}_{i,j} = \frac{\sum_{k=1}^{\min(|bs_i|,|bs_j|)} bs_{i,k} \oplus bs_{j,k}}{\min(|bs_i|,|bs_j|)} \tag{12}$$

Equation (12) gives the expression for Inter-HD for a pair of devices ($i$, $j$). The Thresholding and XMR operations produce concatenated bitstrings of different lengths for each device. The summation is therefore performed over the length of the shorter bitstring. The Inter-HD is computed for all possible 7140 combinations of the 120 bitstrings and averaged. The results are shown by the bar graphs in Figure 13a, which depict Inter-HD values very close to the ideal value of 50%. Closer inspection reveals the values vary from 50.004% to 50.008%.

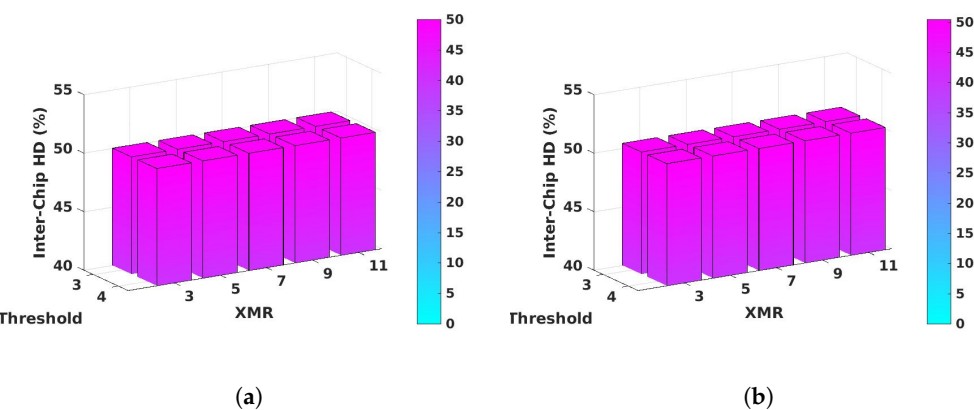

(**a**)             (**b**)

**Figure 13.** Hamming distance results of concatenated bitstrings. (**a**) Inter-HD of un-aligned bitstrings for thresholds 3 and 4. (**b**) Inter-HD of aligned bitstrings for thresholds 3 and 4.

The helper data produced by the Thresholding and XMR operations reveals which bits in the unfiltered bit sequence are used to generate the FSB Response bitstring. An adversary might use this information in an attempt to predict a bitstring outcome by determining if bias exists for bits generated by the same path pairings. Figure 13b gives the Inter-HD results when the analysis is carried out using only bits corresponding to the same path pairing. We refer to the process of selecting strong bits whose positions in the sequence of unfiltered bits is the same as aligning the FSB Response bitstrings. The bar graphs show that Inter-HD values vary between 50.412% and 50.424%.

Although the Inter-HD values are slightly larger than the values shown for the un-aligned analysis, the number of bits that meet the alignment criteria is much smaller in the alignment analysis. For example, the average number of bits used in the un-aligned XMR 11 analysis is 125,000 but reduces to less than 7000 for the aligned analysis. Therefore, the uncertainty created by the smaller sample size, where, for example, only 3.4 bits are used on average per 2048 unfiltered bit sequence, contributes to the observed difference.

### 4.4. FSB Response Bitstring Size Analysis

The average lengths of FSB response bitstrings per iteration of the SiRF PUF algorithm are shown in Figure 14a, which portrays the increased level of filtering that occurs for larger values of the threshold and XMR. The lengths are consistent with those predicted using a uniform random distribution model. For example, the parameters used to generate the bitstrings uniformly distribute the $DVD_{cr}$ over the region between 1 and 9, and $-1$ and $-9$, as shown in Figure 8a. A threshold of 3 discards 2/9ths of the distribution as weak bits, leaving 1593 ($7/9 \times 2048$) bits classified as strong. The number of strong bits is reduced by 25 bits on average to 1569 because $DVD_{cr}$ equal to 0.0 remain at 0.0, i.e., they are not redistributed to one of the regions identified above.

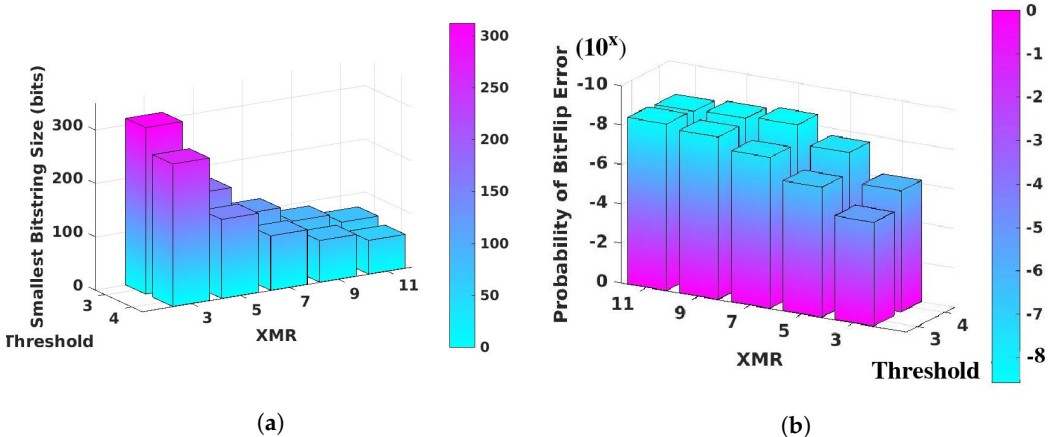

(**a**)             (**b**)

**Figure 14.** Bitstring size and probability-of-failure statistics. (**a**) Smallest bitstring sizes produced by the SiRF PUF algorithm from unfiltered bitstrings of size 2048 bits. (**b**) The probability of failure in regenerating the bitstrings at the fourteen TV corners.

XMR 3 will utilize five strong bits on average to generate one super-strong bit, yielding an average bitstring size of 314 bits per iteration of the SiRF PUF algorithm. In general, the number of strong bits required for each super-strong bit is $XMR \times 2 - 1$. Therefore, XMR 5 requires nine strong bits per super-strong bit, yielding an average FSB response bitstring size of 174. The smallest yield of super-strong FSB response bits is 74 for XMR 11, which requires twenty-one strong bits on average. A similar analysis applies for a threshold of 4 using 6/9ths as the initial fraction. The smallest bar in Figure 14a is 64 for threshold 4, XMR 11.

### 4.5. Reliability Analysis

Reliability, as it relates to PUFs, refers to ability of the PUF to regenerate the response bitstring without errors under adverse environmental conditions. The primary goal of the

GPEVCal, Thresholding and XMR components of the SiRF PUF algorithm, as well as the DV Characterization process, is to improve reliability without resorting to commonly-used error correction tactics.

In this analysis, the concatenated bitstrings generated under nominal conditions, 25 °C, 1.00 V, are paired with the sets of bitstrings generated at the fourteen TV corners for each device. Equation (13) is used to compute the Intra-chip hamming distance (Intra-HD) for each pairing. The tuple $(i, n, j)$ identifies a pairing for device $i$ which pairs the nominal bitstring $n$ with the bitstring generated at TV corner $j$. Note that unlike Inter-chip HD, the expression for Intra-HD simply computes the total number of bit flip errors that occur in the pairing. The probability of a bit flip error reported in the following is computed by dividing the total number of bit flip errors for all devices and TV corners by the total number of bit-pairings inspected.

$$\text{Intra-HD}_{i,n,j} = \sum_{k=1}^{|bs_i|)} bs_{i,n,k} \oplus bs_{i,j,k} \tag{13}$$

The parameters to Thresholding and XMR are tunable, with larger values providing higher reliability. The bar graphs in Figure 14b depict the probability of a bit flip error occurring during FSB Response bitstring regeneration under one (or more) of the TV corners. The values on the z-axis represent the exponents to $10^x$, e.g., a value of $-8$ indicates one bit flip error occurred among the 100 million bits inspected.

The height of the bars for threshold 4, XMRs 7, 9 and 11 vary between $-8.37$ and $-8.58$. In these test cases, no bit flip errors were observed in the analysis of more than 230 million bits. The values reported are computed assuming one bit flip error occurs, i.e., they are computed as the reciprocal of the number of bits inspected. Therefore, the probability of failure for these test cases is unknown, and the results reported actually represent a lower bound on probability of failure. A least one bit flip error occurred for the other threshold-XMR combinations but the probabilities remain below $1/10^6$ for XMRs of 5 and higher for both thresholds. The one-in-a-million threshold is widely recognized as the target reliability standard for PUFs.

The most challenging TV corners for regeneration are 100 °C, 1.05 V and $-40$ °C, 0.95 V, with the former exhibiting more than twice the number of bit flip errors. Figure 15 plots the number of bit flip errors observed across all TV corners for threshold 3 as a function of the XMR level. The worst case TV corners are most noticeable at XMR 3, which shows more than 4500 bit flip errors for TV corner 100 °C, 1.05 V among the 1.2 billion bits inspected. It should be noted that the number of bits inspected decreases as XMR increases, e.g., the numbers for XMRs 5 through 11 are 640, 443, 339 and 274 million, respectively.

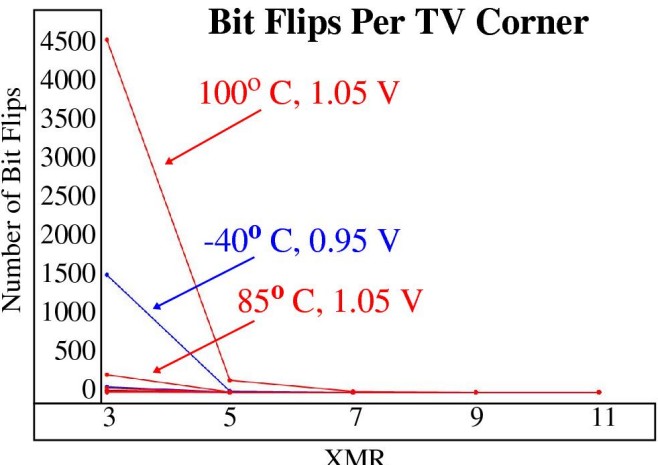

**Figure 15.** Bit flips per TV corner as a function of XMR for bitstrings generated with the threshold set to 3.

Given the FPGAs-under-test are commercial grade, with a specified operating range between 0 °C and 70 °C, the reliability results presented here provide strong evidence that the proposed techniques provide reliability that meets or exceeds the target reliability standard expected for PUFs.

## 5. Implementation Characteristics

Characteristics of the FPGA implementation, performance and challenge-response-pair (CRP) space of the SiRF PUF are presented in this section.

### 5.1. Resource Utilization and Power

The SiRF PUF resource utilization on a Xilinx Zynq 7010 FPGA is given in Table 1 separately for the SiRF engine components and the engineered netlist component. The SiRF engine components are shown as a set of modules (implemented as state machines) on the right side of Figure 2 while the engineered netlist is shown on the left and in Figure 1. The resource utilization is computed with the SiRF engine configured with five security functions, including device and server authentication, session key generation, long-lived key generation and a TRNG. Applications that require only a subset of these functions reduce the resource utilization by approx. 10% for each function removed.

**Table 1.** SiRF PUF resource utilization on the Zynq 7010.

| Resource | SiRF Engine Utilization | Netlist Utilization | Available | Utilization % |
|---|---|---|---|---|
| LUT | 5842 | 796 | 17,600 | 37.72 |
| LUTRAM | 60 | 96 | 6000 | 2.60 |
| FF | 4377 | 32 | 35,200 | 12.53 |
| BRAM | 5 | - | 60 | 8.33 |
| DSP | 2 | - | 80 | 2.50 |
| BUFG | 2 | - | 32 | 6.25 |

The power analysis tool in Xilinx Vivado estimates power consumption at 1.43 W for the Cortex A9 microprocessor (PS side) and 27 mW for the SiRF implementation in the programmable logic (PL side). Although the Zynq 7010 device is one of the smallest SoC-based FPGAs that Xilinx manufacturers, the processing system (PS) side consumes a large fraction of the total power. For battery-powered IoT applications, an ASIC implementation or FPGA-only device provides a better platform from a power perspective.

### 5.2. Performance Analysis

The statistical results presented in the previous sections show that cryptographic quality in the SiRF PUF bitstrings can be achieved for any of the threshold and XMR parameters. This makes it possible to freely choose the parameters to achieve a target reliability standard, while trading off the number of iterations performed by the algorithm to generate a fixed size bitstring, and the corresponding bitstring generation time.

The typical run time for one iteration of the SiRF algorithm for either enrollment or regeneration is approximately 1.4 s. Table 2 gives the total run time to generate a 256-bit encryption key as a function of XMR. The run times include the network delay to retrieve the configuration vectors and SpreadFactors from a server (an operation that only occurs during enrollment where the encryption key is generated for the first time).

**Table 2.** SiRF PUF (re)generation run times to generate a 256-bit encryption key.

| XMR | Run Time (s) | Number of Iterations |
|:---:|:---:|:---:|
| 3 | 1.40 | 1 |
| 5 | 1.50 | 2 |
| 7 | 1.58 | 3 |
| 9 | 1.58 | 3 |
| 11 | 1.67 | 4 |

Note that the path timing operation, which generates the $DV_R$ and $DV_F$, is run during the first iteration only. Subsequent iterations change the LFSR seed input parameters to the Difference module, to create a new set of DVD. The largest portion of the run time is associated with path timing operation, and therefore, XMR setting that require multiple iterations add only small increases, on order of 100 milliseconds, to the overall run time.

*5.3. Challenge-Response-Pair (CRP) Analysis*

The engineered netlist with an array of $3 \times 8$ modules as shown in Figure 1 possesses $2^{24}$ distinct rise and fall paths. As discussed in Section 3.4.3, multiple compatibility sets of cardinality of approx. 20,000 paths can be selected, yielding approx. 838 distinct, non-overlapping compatibility sets (all paths are compatible with some subset of paths from the initial set). Assuming each compatibility set consists of 10,000 $DV_R$ and 10,000 $DV_F$, the number of distinct path delay differences (DVD) is equal to 100,000,000. Therefore, the total number of unique path combinations (and corresponding bits) across all 838 compatibility sets is approx. 84 billion ($2^{36}$). The range constant, $r_c$ and SpreadFactors increase the number of bits by a factor of 16 to approx. $2^{40}$.

Bear in mind that adding another row of modules to the block diagram shown in Figure 1 increases the number of distinct rise and fall paths exponentially. Each additional row adds a factor of 64 to the number of paths for each output. With four rows and 32 outputs, this generates $64^4 \times 32 = 2^{29}$ physical paths, each of which can be tested with a rising or falling transition, for a total of $2^{30}$ paths. This increases the total number of bits to $2^{46}$. Moreover, the transition direction challenge bit ($TC_x$) that is associated with each row allows different segments of each path to be tested with combinations of rising and falling transitions, increasing the number of bits even further.

## 6. Conclusions and Future Work

A physical unclonable function called the shift-register, reconvergent-fanout (SiRF) PUF is proposed, and test results are presented that demonstrate the statistical quality of the response bitstrings. An algorithm is described that post-processes digitized path delays measured through an engineered netlist of shift-registers and logic gates. The algorithm reduces undesirable variations in path delays introduced by chip-to-chip process variations and adverse environmental conditions. Other components of the algorithm improve the uniqueness and randomness statistical characteristics of the generated bitstrings.

The SiRF PUF is implemented as a set of state machines in the programmable logic of a Xilinx FPGA. The results presented in this paper are derived from the data collected in a set of temperature-voltage experiments carried out on 120 SiRF PUF instances. The reliability of the SiRF PUF is evaluated using test data collected across a set of TV corners defined by the extended industrial range standard.

The results from statistical tests which measure the uniqueness, randomness and reliability of the response bitstrings show the bitstrings meet cryptographic quality and reliability standards, and the bitstrings are therefore suitable for encryption and authentication applications. On-going work is focused on the integration of the SiRF PUF into challenge-response-based secure boot and communication protocols, which leverage the unique capabilities of the PUF including the exponentially large number of response bit-

strings that are available. The development of a hardware-based secure enclave is also part of on-going work which combines the SiRF PUF with a set of hardware-instantiated cryptographic functions.

**Funding:** This research received no external funding.

**Institutional Review Board Statement:** Not applicable

**Informed Consent Statement:** Not applicable

**Data Availability Statement:** Not applicable

**Conflicts of Interest:** The author declares no conflict of interest.

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
