# Peer review of "Shift Register, Reconvergent-Fanout (SiRF) PUF Implementation on an FPGA"

_cryptography, doi:10.3390/cryptography6040059_

Round 1

Reviewer 1 Report

Two things are not clear: expected chip area & chip power consumption. Normally we have a cost for any sort of security improvement, so this cost (defined in terms of hdw penalty) is not clearly emphasized/described. 

Author Response

Comment 1.1 Two things are not clear: expected chip area & chip power consumption. Normally we have a cost for any sort of security improvement, so this cost (defined in terms of hdw penalty) is not clearly emphasized/described.

Reply: I have added a new section 5.0 called Implementation characteristics that provides information about the resource utilization and power consumption on FPGAs

Reviewer 2 Report

I very much liked reading about new ways of retrieving stable PUFs. The article is scientifically sound, it describes the solution in details and following results prove that the idea works. It was a little difficult to understand the PUF generation based on Figure 1 (quite a lot of details), however, with the help of Figure 2 all became clear enough.

I do not have major concerns, just small remarks that may help to improve the article.

NIST test suite would not be my first choice to assess PUF quality (of course after the basics: randomness, uniqueness and reliability). Nowadays, authors prove the PUF ideas with add attacks based on machine learning. I recommend it, but I don’t fell that it is mandatory here.   

I understand that the solution may not win the cup, however, it is a good practice to compare a few solutions at the end of the article in regard to basic parameters (e.g., implementation costs, speed, quality, C-R vectors spaces etc.).

Small editorial corrections would be appreciated – e.g.:

Line 535 – capital C for Celsius

Lines 520, 549, … – hyphens instead of minuses

Lines 521, 526, … – probably not the best idea to use asterisk for multiplication outside simple text editors or programming (just my opinion, others say the asterisk is used for convolution product)

Page 20 – never use the default math italic font for multi-letter words (e.g. DVD) – it is designed to make consecutive letters not look like a word, but as a product of variables (D times V times D).  $\mathit{DVD}$ will fix the problem.

In conclusion, the article is very much suitable for the journal. 

Author Response

Comment 2.1 I very much liked reading about new ways of retrieving stable PUFs. The article is scientifically sound, it describes the solution in details and following results prove that the idea works. It was a little difficult to understand the PUF generation based on Figure 1 (quite a lot of details), however, with the help of Figure 2 all became clear enough.

Reply: Thank you.

Comment 2.2 I do not have major concerns, just small remarks that may help to improve the article. NIST test suite would not be my first choice to assess PUF quality (of course after the basics: randomness, uniqueness and reliability). Nowadays, authors prove the PUF ideas with add attacks
based on machine learning. I recommend it, but I don’t fell that it is mandatory here.

Reply: I agree that ML is important to assess for strong PUFs but the application we are pitching in this paper is secure boot (Section 3.4.2), where the keys are not exposed so ML attacks are not possible. We have another paper on using the SiRF PUF for authentication where we plan to address the ML attack issues.

Comment 2.3 I understand that the solution may not win the cup, however, it is a good practice to compare a few solutions at the end of the article in regard to basic parameters (e.g., implementation costs, speed, quality, C-R vectors spaces etc.).

Reply: I added a new section 5.0 called Implementation Characteristics that provides information about
the resource utilization, power, speed, CRP space, etc.

Comment 2.4 Small editorial corrections would be appreciated – e.g.:
Line 535 – capital C for Celsius
Lines 520, 549, . . . – hyphens instead of minuses
Lines 521, 526, . . . – probably not the best idea to use asterisk for multiplication outside simple text editors or programming (just my opinion, others say the asterisk is used for convolution product)
Page 20 – never use the default math italic font for multi-letter words (e.g. DVD) – it is designed to make consecutive letters not look like a word, but as a product of variables (D times V times D). DVD will fix the problem.

Reply: Thank you – all fixed. Decided to drop italic for DV, DVD, etc, and added DVDcr, etc.

Comment 2.5 In conclusion, the article is very much suitable for the journal.

Reply: Thank you.

Reviewer 3 Report

In this work the shift-register, reconvergent-fanout (SiRF) PUF has been proposed as a solution to enhance the uniqueness of the response and to increase reliability under environmental variations. Furthermore, an algorithm to increase bit-stream quality and hide systematic variations is proposed, and measuremnts results are presented. Validation through a subset of NIST tests demonstrates the statistical quality of the response and the effectiveness of the approach.

 Even though the principle presented in this work is good and relevant, some major improvements are necessary to consider this work for publication.

 Concerns are as follows:

 Introduction:

â—¦      In chapter 1.1 when authors presented the background of PUFs, some important architectures presented in literature as lightweight cryptographic solution are missing and should be inserted to improve this section, for example:

1.     Marchand, Cédric, et al. "Implementation and Characterization of a Physical Unclonable Function for IoT: A Case Study With the TERO-PUF." IEEE Trans. Comput. Aided Des. Integr. Circuits Syst., vol. 37, no. 1, 9 May. 2017, pp. 97-109.

2.     Bossuet, Lilian, et al. "A PUF Based on a Transient Effect Ring Oscillator and Insensitive to Locking Phenomenon." IEEE Trans. Emerging Top. Comput., vol. 2, no. 1, 28 Oct. 2013, pp. 30-36.

3.     Della Sala, Riccardo, et al. "A Novel Ultra-Compact FPGA PUF: The DD-PUF." Cryptography, vol. 5, no. 3, 8 Sept. 2021, p. 23.

4.     Gu, C.; Hanley, N.; O’Neill, M. “Improved reliability of FPGA-based PUF identification generator design”. ACM Trans. Reconfigurable Technol. Syst. (TRETS) 2017, 10, 1–23.

5.     Gu, C.; Chang, C.H.; Liu, W.; Hanley, N.; Miskelly, J.; O’Neill, M. “A large-scale comprehensive evaluation of single-slice ring oscillator and PicoPUF bit cells on 28-nm Xilinx FPGAs”. J. Cryptogr. Eng. 2020, 11, 1–12.

6.     Della Sala, Riccardo, et al. "A Lightweight FPGA Compatible Weak-PUF Primitive Based on XOR Gates." IEEE Trans. Circuits Syst. II, vol. 69, no. 6, 4 Mar. 2022, pp. 2972-6.

7.     Maes, R.; Tuyls, P.; Verbauwhede, I. “Intrinsic PUFs from flip-flops on reconfigurable devices”. In Proceedings of the 3rd Benelux Workshop on Information and System Security (WISSec 2008), Eindhoven, The Netherlands, 13–14 November 2008; Volume 17, p. 2008.

8.     Huang, Zhao, et al. "RPPUF: An Ultra-Lightweight Reconfigurable Pico-Physically Unclonable Function for Resource-Constrained IoT Devices." Electronics, vol. 10, no. 23, 5 Dec. 2021, p. 3039.

9.     Kumar, S.S.; Guajardo, J.; Maes, R.; Schrijen, G.J.; Tuyls, P. “The Butterfly PUF: Protecting IP on every FPGA”. In Proceedings of the IEEE International Workshop on Hardware-Oriented Security and Trust, HOST 2008, Anaheim, CA, USA, 9 June 2008; pp. 67–70.

10.  Gu, C.; Murphy, J.; O’Neill, M. “A unique and robust single slice FPGA identification generator”. In Proceedings of the 2014 IEEE International Symposium on Circuits and Systems (ISCAS), Melbourne, VIC, Australia, 1–5 June 2014; pp. 1223–1226.

  • SiRF Design:

â—¦      Please, export figures 1 and 2 in high resoultion, since it appear out of focus;

â—¦      Also Fig 3 appears out of focus, please export it in high resoultion.

â—¦      Since one of the most important aspect in delay based PUFs is the symmetry in routing connections and also the symmetry in placing, authors have to insert a further section in SiRF Design, describing in detail the routing and placing strategies as in [1,2]:

1.     Gu, C.; Chang, C.H.; Liu, W.; Hanley, N.; Miskelly, J.; O’Neill, M. “A large-scale comprehensive evaluation of single-slice ring oscillator and PicoPUF bit cells on 28-nm Xilinx FPGAs”. J. Cryptogr. Eng. 2020, 11, 1–12.

2.     Marchand, Cédric, et al. "Implementation and Characterization of a Physical Unclonable Function for IoT: A Case Study With the TERO-PUF." IEEE Trans. Comput. Aided Des. Integr. Circuits Syst., vol. 37, no. 1, 9 May. 2017, pp. 97-109.

Indeed, it is not clear at all, if the focus of the paper is on an unbiased and also place dependent (i.e. with very low uniqueness or intra-HD performance) architecture, or on an enhancement of an already biased architecture. Please, rearrange this section by considering these comments in order to make the focus of the paper more clear. As a matter of fact, if a balanced routing strategy was adopted, mismatch contributions of nominally identical path delays will enhance the uniqueness of the architecture. It seems that this work is focused on the opposite strategy, that is an unbalanced routing which with some algorithm technique is guaranteed as symmetric as possible, please clarify this point and compare this design strategy with those adopted in:

1.     Della Sala, Riccardo, et al. "A Novel Ultra-Compact FPGA PUF: The DD-PUF." Cryptography, vol. 5, no. 3, 8 Sept. 2021, p. 23.

2.     Gu, C.; Hanley, N.; O’Neill, M. “Improved reliability of FPGA-based PUF identification generator design”. ACM Trans. Reconfigurable Technol. Syst. (TRETS) 2017, 10, 1–23.

3.     Gu, C.; Chang, C.H.; Liu, W.; Hanley, N.; Miskelly, J.; O’Neill, M. “A large-scale comprehensive evaluation of single-slice ring oscillator and PicoPUF bit cells on 28-nm Xilinx FPGAs”. J. Cryptogr. Eng. 2020, 11, 1–12.

4.     Della Sala, Riccardo, et al. "A Lightweight FPGA Compatible Weak-PUF Primitive Based on XOR Gates." IEEE Trans. Circuits Syst. II, vol. 69, no. 6, 4 Mar. 2022, pp. 2972-6.

1.     Bossuet, Lilian, et al. "A PUF Based on a Transient Effect Ring Oscillator and Insensitive to Locking Phenomenon." IEEE Trans. Emerging Top. Comput., vol. 2, no. 1, 28 Oct. 2013, pp. 30-36.

  • Comparison with state of the art:

In this work a comparison with state of the art is missing, please insert a comparison table like the one reported in:

1.     Huang, Zhao, et al. "RPPUF: An Ultra-Lightweight Reconfigurable Pico-Physically Unclonable Function for Resource-Constrained IoT Devices." Electronics, vol. 10, no. 23, 5 Dec. 2021, p. 3039, doi:10.3390/electronics10233039.

2.     Della Sala, Riccardo, et al. "A Lightweight FPGA Compatible Weak-PUF Primitive Based on XOR Gates." IEEE Trans. Circuits Syst. II, vol. 69, no. 6, 4 Mar. 2022, pp. 2972-6.

Author Response

Comment 3.1 In this work the shift-register, reconvergent-fanout (SiRF) PUF has been proposed as a solution to enhance the uniqueness of the response and to increase reliability un-
der environmental variations. Furthermore, an algorithm to increase bit-stream quality and hide systematic variations is proposed, and measuremnts results are presented. Validation through a subset of NIST tests demonstrates the statistical quality of the response and the effectiveness of
the approach. Even though the principle presented in this work is good and relevant, some major improvements
are necessary to consider this work for publication.
Concerns are as follows:
Introduction:
â—¦ In chapter 1.1 when authors presented the background of PUFs, some important architectures presented in literature as lightweight cryptographic solution are missing and should be inserted to improve this section, for example:

(recommended reference additions deleted)

Reply: I have added references and short descriptions for most of the papers identified above to the background section.

Comment 3.2 SiRF Design:
â—¦ Please, export figures 1 and 2 in high resoultion, since it appear out of focus;
â—¦ Also Fig 3 appears out of focus, please export it in high resoultion.
â—¦ Since one of the most important aspect in delay based PUFs is the symmetry in routing connections and also the symmetry in placing, authors have to insert a further section in SiRF
Design, describing in detail the routing and placing strategies as in [1,2]:

(recommended reference additions deleted)

Reply: I re-imported Figs 1, 2, 3, 4, 5, 8 and 10 to remove the fuzziness. There is no fixed placement
or routing needed for the SiRF PUF. I have emphasized
this point at the end of Section 2.0, and in other (highlighted in blue) places in the revised paper.

Comment 3.3 Indeed, it is not clear at all, if the focus of the paper is on an unbiased and also place dependent (i.e. with very low uniqueness or intra-HD performance) architecture, or on an enhancement of an already biased architecture. Please, rearrange this section by considering these comments in order to make the focus of the paper more clear. As a matter of fact, if a balanced routing strategy was adopted, mismatch contributions of nominally identical path delays will enhance the uniqueness of the architecture. It seems that this work is focused on the opposite strategy, that is an unbalanced routing which with some algorithm technique is guaranteed as
symmetric as possible, please clarify this point and compare this design strategy with those adopted in

(recommended reference additions deleted)

Reply: It is true that the SiRF PUF takes the opposite approach to identically-designed subcircuit-
based PUFs. In addition to the modification to Section 2.0, I have also added (and highlighted) some
text in Section 3.3 that makes this distinction clear.

Comparison with state of the art:
In this work a comparison with state of the art is missing, please insert a comparison table like the one reported in:

(recommended reference additions deleted)

Reply: In general, it is difficult to compare weak PUFs and strong PUFs. The former is designed to generate only a single key or a small number of them and so resource utilization is always very different. I have added Section 5.0 to elaborate on the details of the SiRF PUF implementation characteristics.